# Are Classification Deep Neural Networks Good for Blind Image Watermarking?

**DOI:** 10.3390/e22020198

**Published:** 2020-02-08

**Authors:** Vedran Vukotić, Vivien Chappelier, Teddy Furon

**Affiliations:** 1Lamark, 35000 Rennes, France; vedran.vukotic@imatag.com (V.V.); vivien.chappelier@lamark.fr (V.C.); 2INRIA, CNRS, IRISA, University of Rennes, 35000 Rennes, France

**Keywords:** digital watermarking, Deep Learning, feature extraction

## Abstract

Image watermarking is usually decomposed into three steps: (i) a feature vector is extracted from an image; (ii) it is modified to embed the watermark; (iii) and it is projected back into the image space while avoiding the creation of visual artefacts. This feature extraction is usually based on a classical image representation given by the Discrete Wavelet Transform or the Discrete Cosine Transform for instance. These transformations require very accurate synchronisation between the embedding and the detection and usually rely on various registration mechanisms for that purpose. This paper investigates a new family of transformation based on Deep Neural Networks trained with supervision for a classification task. Motivations come from the Computer Vision literature, which has demonstrated the robustness of these features against light geometric distortions. Also, adversarial sample literature provides means to implement the inverse transform needed in the third step above mentioned. As far as zero-bit watermarking is concerned, this paper shows that this approach is feasible as it yields a good quality of the watermarked images and an intrinsic robustness. We also tests more advanced tools from Computer Vision such as aggregation schemes with weak geometry and retraining with a dataset augmented with classical image processing attacks.

## 1. Introduction

Deep Learning (DL) has completely revolutionized the field of Computer Vision. It started with image classification [1] where a DL network is trained to recognize classes of images and it is now spreading to any task of Computer Vision—object recognition [2], instance search [3], localization [4], similar image retrieval [5,6].

The transfer property is underlying this versatility—a DL network trained on some supervised task (e.g., classification) has been shown to perform well on other applications, even non-supervised tasks like similarity search.

A DL network is comprised of several layers. Supervised training learns the best parameters of each layer in order to minimize a loss function counting the prediction errors over an annotated image set. While the last layer of a classifier network provides a vector of probabilities that the input image belongs to the classes, the internal auxiliary data (the output of the previous layers) also happen to contain very relevant information about the input image. Other tasks can indeed tap these data. Many Computer Vision works take a trained classification network off the shelf, keep the first layers as is and only re-train the deepest layers for their specific task [5].

Following this trend, state-of-the-art image search algorithms tap the output of an internal layer and use them to derive a global descriptor of the image [5,6]. This way, finding similar images boils down to finding close vectors in a Euclidean high dimensional space, where efficient fast search engines exist. It has been shown that these global descriptors are compact and discriminative while at the same time robust to valuemetric (e.g., JPEG, noise, filtering) and light geometric (e.g., cropping, rotation, scaling) distortions [7].

### 1.1. Problem Formulation

This paper investigates whether this approach is also suitable for zero-bit watermarking, with the hope of benefiting from this apparent robustness. The layers of a DL network play the role of the extraction function yielding a vector to be watermarked. This raises the following challenges:How to invert this highly non-linear extraction function? Once the feature vector is watermarked, how to map it back into the image space to create an image whose feature vector equals a given target vector?How to take into account a perceptual model ensuring the quality of the watermarked image?How to guarantee a required false positive level as there is no probabilistic modeling of these features?Which network architecture provides the best extraction function? Is it possible to robustify this extraction by fine-tuning the underlying network?

### 1.2. Prior Works

The main trend dealing with DL and watermarking is actually the protection of DL networks. It aims at proving the ownership of a network by embedding a watermark into the learned parameters of the model [8,9].

The connection between machine learning and *image* watermarking is surprisingly old. References [10,11] protect images with a classical watermarking technique and a neural network is only used at the decoding side in place of a maximum likelihood decoder based on a statistical model, which may not be accurate. J. Stolarek and P. Lipiński learn a transformation [12] as proposed in this paper. However, this transform is dedicated to one specific host image. References [13,14] have a similar position than ours in the sense that they use a single network at the detection side. Reference [14] needs to train another network for embedding, whereas [13] uses back-propagation as we do (see Section 4.3). There are also shortcomings that watermarking experts would point out. These are multi-bit watermarking schemes that fail guaranteeing a probability of false alarm. Indeed, Reference [14] briefly looks at turning their multi-bit embedding into a zero-bit scheme. They report a detection rate lower than 10% for a JPEG compression with quality factor Q≤90 (see [14], Figure 11-combined). There is no secret key preventing renewability when compromised. There is no simple control of the embedding distortion. The embedding distortion reported in both papers is indeed quite high with an average PSNR around 35 dB. They are not robust to geometric transformations (unless a registration is performed first [13]).

As a related work, we mention Reference [15] which makes a very good comparison between attacks in watermarking and attacks on DL networks (*a.k.a.* adversarial sample—see Section 3.2).

This article is the journal version of the conference paper [16] proposing improvements to robustify the feature vector extraction. This is achieved by using aggregation schemes with weak geometry (RMAC and WELDON—see Section 3.1) and by an adversarial retraining of the network (see Section 4.4). This last idea is coming from References [13,14].

The contribution of the study is to assess the relevance of Deep Neural Networks as a feature extraction when watermarking requirements are prioritized: embedding and detection use a secret key which is easily renewed if compromised, the embedding distortion is under control, the probability of false alarm is guaranteed and the test images are large.

### 1.3. Structure of the Paper

Section 2 presents the state-of-the-art in zero-bit watermarking when combined with a linear extraction function. Section 3 presents elements of DL networks with a focus on adversarial examples. Section 4 details our new watermarking scheme and a re-training process to increase the robustness by strengthening the invariance of the network. Section 5 reports our experimental work following by a discussion in Section 6.

## 2. Zero-Bit Watermarking

Our scheme belongs to the TEMIT approach (a wording coined by Ton Kalker.): Transform, EMbed, Inverse Transform. We denote by Tr(·) the transformation extracting a feature vector, so-called the host signal $x_{o}=\mathrm{Tr}\left(I_{o}\right)$ of dimension *n*, from an image $I_{o}$. The embedding modifies it into $x_{m}=\mathrm{Emb}\left(x_{o}\right)$. This function implicitly depends on a secret key. For simplicity, the paper focuses on a zero-bit watermarking scheme [17,18,19]: we hide the presence of a secret mark, not a message. The watermarked image is given by $I_{m}=I_{o}+{\mathrm{Tr}}^{-1}\left(x_{m}-x_{o}\right)$, where ${\mathrm{Tr}}^{-1}\left(\cdot \right)$ is the inverse transform function.

At the detection side, the image under scrutiny is $I_{a}$ with $x_{a}=\mathrm{Tr}\left(I_{a}\right)$. It is deemed as a watermarked image if $x_{a}$ belongs to the acceptance region $D\subset R^{n}$. The use of a secret key is implicit in these notations.

### 2.1. The Hypercone Detector

This paper focuses on one specific zero-bit watermarking scheme: the hypercone detector [17]. This scheme is provably good [19] and the detector is blind and oblivious to the watermark, host and noise powers.

The acceptance region D is a dual hypercone of axis $a\in R^{n}$ (∥a∥=1) and half angle 0<θ<π/2 defined as:(1)$$D:=\{x\in R^{n}:|x^{⊤}a|>∥x∥\cos\left(\theta \right)\}.$$

Vector a indeed plays the role of the secret key.

We now define the following function $R\left(\cdot \right):R^{n}\mapsto R$
(2)$$x\to R\left(x\right)={\left(x^{⊤}a\right)}^{2}\rho \left(\theta \right)-{∥x∥}^{2},$$
with $\rho \left(\theta \right):=(\tan^{2}\left(\theta \right)+1)$. This quantity is negative when x∉D, null when x lies on the boundary of D and positive inside the dual hypercone. Indeed, when R(x)>0, this quantity is the amount of power of Gaussian white noise to be added onto x in order to push it outside the hypercone with high probability [19].

This gives a rationale for watermarking vectors at the embedding side. We push the host vector $x_{o}$ to a location $x_{m}$ deep inside D s.t. $R(x_{m})$ is as big as possible. This provably increases the robustness against a white Gaussian noise addition in the feature space.

### 2.2. Linear and Invertible Extraction Function

The hypercone detector is at the core of some image watermarking techniques [18] where the extraction function is (almost) a Parseval tight frame *T* (e.g., selection of some DCT or DWT coefficients). Once the embedding modifies $x_{o}$ in $x_{m}$, the watermarked image is obtained as
(3)$$I_{m}=I_{o}+T^{⊤}\left(x_{m}-x_{o}\right).$$

A constraint on the image Euclidean distance $∥I_{m}-I_{o}{∥}_{2}\leq C$ (e.g., expressed in terms of MSE or PSNR) is ensured if $∥x_{m}-x_{o}{∥}_{2}\leq C$ because Parseval frame preserves Euclidean distance. In the feature domain, the embedding then amounts to solve the following problem:(4)$$x_{m}=\arg\max_{x:∥x-x_{o}{∥}_{2}\leq C}R\left(x\right).$$

It is known that $x_{m}$ belongs to the 2D-hyperplane containing the origin $x_{o}$ and the axis of the hypercone a [17,20], so that (Equation 4) boils down to a line search over angle β∈[0,π/2] defining:(5)$$x=x_{o}+C\cos\left(\beta \right)u-C\sin\left(\beta \right)v,$$
with (u,v) a basis of this hyperplane:(6)$$u:=\mathrm{sign}\left(x_{o}^{⊤}a\right)a,\;v:=\frac{x_{o}-\left(x_{o}^{⊤}u\right)u}{∥x_{o}-\left(x_{o}^{⊤}u\right)u∥}.$$

Vectors $x_{o}$ and a are statistically independent, so that for large *n*, $x_{o}^{⊤}a\approx 0$. In that case, the close form solution is [19]:(7)$$\beta^{☆}=\arccos\sqrt{1-∥x_{o}{∥}^{2}C^{-2}\cos^{4}\theta },$$
if $C\geq ∥x_{o}∥\cos^{2}\theta$, otherwise $x_{o}$ cannot be watermarked (i.e., pushed into region D) because *C* is too small.

The embedding is thus given by a close form solution or a simple line search over β. This is thanks to an invertible extraction function Tr(·) preserving the Euclidean distance.

## 3. Deep Learning Feature Extraction

This section considers a neural network used as an image classifier over $n_{c}$ classes. It is denoted by function F that computes a vector from an image: $p=F\left(I\right)\in {\left[0,1\right]}^{n_{c}}$. Once the parameters of the classifier have been learned from a training set, the network predicts classes of images. The output p is a vector of estimated probabilities $p_{k}=\hat{P}\left(k|I\right)$ that content I belongs to class *k*. In the end, the class of the query image is given by
(8)$$C\left(I\right)=\arg\max_{1\leq k\leq n_{c}}p_{k}.$$

### 3.1. Architecture

The function F is decomposed as F=S∘N, where S is the softmax function ensuring that p is a probability distribution (i.e., $\sum_{k=1}^{n_{c}}p_{k}=1$ and $0\leq p_{k}\leq 1$, $\forall k\in \{1,\ldots ,n_{c}\}$). Function N is the network per se, a composition of $\ell_{C}$ convolutional layers and $\ell_{F}$ fully connected layers.
(9)$$N=L_{\ell_{F}}^{F}\circ L_{\ell_{F}-1}^{F}\circ \ldots L_{1}^{F}\circ V\circ L_{\ell_{C}}^{C}\circ L_{\ell_{C}-1}^{C}\circ \ldots L_{1}^{C}.$$

Each layer is a non-linear process as it encompasses an activation function such as the well-known ReLU. Function V is a layer flattening the 3D activation map of the last convolutional layer into a vector suitable for the first fully connected layer.

The convolutional layers output data living in very high-dimensional spaces that are not practical for direct representation. The output of a convolutional layer is a three-dimensional tensor of size W×H×K, where *K* is the number of feature maps and (H,W) are the dimensions of each feature map. The total dimensionality is in an order of tens or hundreds of thousands, especially in higher layers of deeper network architectures. Our first approach watermarks the output of one layer. If it is a convolutional layer, the flattening function V produces a vector. It is used directly or by performing a dimensionality reduction [16]. In this work, a second approach considers a non-linear pooling strategy. It aggregates the 3D tensor in a specific way for the retention of localization information. It has been shown that geometry even weakly preserved improves a lot the transferability of a classification network to tasks like image search [21]. This work evaluates two aggregation methods in the context of watermarking with neural architectures—RMAC [21] and WELDON [22]. That aggregation is not part of the classification network (Equation 9). It is just added after a convolutional layer for the purpose of watermarking only.

RMAC, namely Regional Maximum Activation of Convolutions, is a multi-scale aggregation method. It retains locality information by utilizing region-specific feature vectors for multiple, overlapping, multi-scale regions. It was developed primarily for content based image retrieval (CBIR) applications of convolutional neural networks. Let $\chi_{i}\left(c\right)$ represent the response of feature i∈[K] at coordinate c∈[W]×[H] (Notation [n] means the set of integers {1,2,…,n}.). Let $f_{R,i}=\max_{c\in R}\chi_{i}\left(c\right)$ be the maximum activation of the $i^{th}$ feature over a valid spatial region R⊂[W]×[H]. A region feature vector is then defined as $f_{R}={\left[\begin{array}{ccccc} f_{R,1} & \cdots & f_{R,i} & \cdots & f_{R,K} \end{array}\right]}^{T}$. Then $\ell_{2}$ normalization and PCA-whitening is applied to each region feature vector. The global (for the whole image I) RMAC descriptor is the sum of the region feature vectors followed by a final $\ell_{2}$ normalization. Its dimension is thus *K*.
(10)$$E_{\mathrm{RMAC}}\left(I\right)\propto \left(\sum_{R}W_{R}^{-1/2}f_{R}/∥f_{R}∥\right)\;s.t.\;∥E_{\mathrm{RMAC}}\left(I\right)∥=1.$$

As RMAC regions are defined at different scales, the first level consists of the whole feature map. The second level consists of 4 regions. The third level consists of 9 regions where each overlap at least 40% with the regions of the previous level and so on. The regions of the first 3 levels of RMAC aggregation are shown, for a feature map of size 14×14×n in Figure 1. In the experimental section, we used an RMAC layer consisting of 5 scale levels. A higher number of scales would not be possible given the size of the feature maps and smaller number of levels were less performing.

WELDON, namely WEakly supervised Learning of Deep cOnvolutional neural Networks, is another method of regional pooling incorporating negative evidences. For a given feature i∈[K], it looks for the *k* coordinates having the largest activation. This is written as:(11)$$s_{i}^{\mathrm{top}}=\sum_{j=1}^{k}\chi_{i}\left(c_{j}\right),$$
where coordinates $\left\{c_{j}\right\}\subset \left[W\right]\times \left[H\right]$ are ordered s.t. $\chi_{i}\left(c_{1}\right)>\chi_{i}\left(c_{2}\right)>\ldots >\chi_{i}\left(c_{WH}\right)$. In the same way, it sums the *m* lowest responses:(12)$$s_{i}^{\mathrm{low}}=\sum_{j=1}^{m}\chi_{i}\left(c_{WH+1-j}\right).$$

The authors state that introducing this kind of negative instances (regions that best supports the absence of a class) improves the avoidance of local minima and improves the overall classification results. The final descriptor for image I has size *K* and is obtained by summing the two regional scores:(13)$$E_{\mathrm{WELDON}}\left(I\right)=s^{\mathrm{top}}+s^{\mathrm{low}}.$$

### 3.2. Adversarial Samples

Adversarial sampling is a recent trend showing that DL classifiers can easily be fooled [23,24]. The attack forges an image $I_{a}$ visually similar to an original image $I_{o}$ but with a different prediction $C\left(I_{a}\right)\neq C\left(I_{o}\right)$.

In a targeted attack, a specific class is given, say $k_{t}$, together with an original image $I_{o}$ not belonging to this class. The attack finds $I_{a}$ as close as possible to $I_{o}$ s.t. $C\left(I_{a}\right)=k_{t}$:(14)$$I_{a}=\arg\min_{I:C\left(I\right)=k_{t}}\mathrm{Dist}\left(I,I_{o}\right),$$
with Dist a measure of distortion (usually, the $L_{\ell }$-norm of $\left(I-I_{o}\right)$ with ℓ∈{0,1,2,+∞}). In a white box scenario, the attacker knows the network architecture and parameters.

Problem (Equation 14) is impossible to be solved as the region of images classified as $k_{t}$ is unknown. It is replaced by
(15)$$\begin{array}{ccc} I_{a} & = & \arg\min J_{\lambda }\left(I\right) \end{array}$$
(16)$$\begin{array}{ccc} J_{\lambda }\left(I\right) & := & \mathrm{Loss}\left(I,k_{t}\right)+\lambda .\mathrm{Dist}\left(I,I_{o}\right). \end{array}$$
where $\lambda \in R^{+}$ and Loss is a loss function measuring how far I is from being classified as class $k_{t}$. For instance, the loss is a divergence (e.g., Cross-Entropy) between the predicted probability distribution F(I) and the targeted one-shot vector $p^{☆}$, i.e., $p_{k}^{☆}=\delta_{k_{t}}\left(k\right)$, or more simply:(17)$$\mathrm{Loss}\left(I,k_{t}\right)={\left|\max_{k\neq k_{t}}\left(F{\left(I\right)}_{k}\right)-F{\left(I\right)}_{k_{t}}\right|}_{+},$$
with ${\left|a\right|}_{+}=a$ if a>0 and 0 otherwise. This way
(18)$$\begin{array}{ccc} C\left(I\right)=k_{t} & \Leftrightarrow & \mathrm{Loss}(I,k_{t})=0, \end{array}$$
(19)$$\begin{array}{ccc} C\left(I_{o}\right)\neq k_{t} & \Leftrightarrow & \mathrm{Loss}(I_{o},k_{t})>0. \end{array}$$

### 3.3. Practical Solutions

The minimization problem (Equation 15) implicitly uses the domain of images of the same size h×l and same number of channels *c* as the original image I, say ${\left\{0,1,\ldots ,255\right\}}^{h\times l\times c}$. A relaxation looks for a solution over the continuous set ${\left[0,255\right]}^{h\times l\times c}$. Quantization onto integers is applied on the continuous solution to obtain an image. This box-constrained optimization often uses a differentiable mapper m which is a monotonic bijection from R to [0,255], e.g., m(x)=255(tanh(x)+1)/2 as in Reference [24]. This changes the box-constrained optimization problem (Equation 14) into an unconstrained minimization of $J_{\lambda }\left(m\left(X\right)\right)$ over $X\in R^{h\times l\times c}$.

Practical attacks are mostly based on a gradient descent [25,26]. The computation of the gradient of the objective function is not so difficult because one can back-propagate the computation of the gradient through the layers of the network. The attack initializes a random starting point $X^{\left(0\right)}=m^{-1}\left(I_{o}\right)$ and iteratively computes
(20)$$X^{\left(i+1\right)}=X^{\left(i\right)}-\eta \nabla J_{\lambda }\left(m\left(X^{\left(i\right)}\right)\right).$$

It stops when the objective function no longer decreases substantially. This gradient descent goes down to a local minimum when converging. Several rounds with different initialization are competing. Advanced numerical solvers have been also used in the literature, for example, ADAM [24]. The main difficulty is to set λ to a well chosen constant. Theoretically, there exists a range of λ values where the solution of (Equation 15) coincides with the solution of (Equation 14). In practice, this range is unknown and we apply a line search over λ on top of the gradient descent.

Simpler attacks start with a given distortion budget, $\mathrm{Dist}(I_{a},I_{o})\leq C$, and set λ=0. The gradient descent starts at the original image, that is, $X^{\left(0\right)}=m^{-1}\left(I_{o}\right)$ and iterates until either $C\left(m\left(X^{\left(j\right)}\right)\right)=k_{t}$ or $\mathrm{Dist}(m\left(X^{\left(j\right)}\right),I_{o})>C$. In the latter case, the attack failed finding an adversarial sample within the distortion budget.

## 4. Application to Zero-Bit Watermarking

The key idea of this paper is (i) to use the network N or part of it (i.e., the first layers) as the extraction function Tr(·), (ii) to modify the extracted vector as explained in Section 2, (iii) to use an adversarial sample mechanism of Section 3 as ${\mathrm{Tr}}^{-1}\left(\cdot \right)$ in order to put back the marked vector into the image. This raises several issues cleared in this section.

### 4.1. Need of a Locality Transform

Our proposed system can function with any neural architecture. Denote by E the set of the first layers and the aggregation (be it the flattening function, RMAC, or WELDON) used for extracting an embedding of an image: $e=E\left(I\right)\in R^{K}$. Yet, the representational coordinate system of the embedding space does not depict the feasible locations (i.e., the typical values of these extracted vectors). A neural network never provides a zero-mean isotropic embedding e. The usual culprit being the asymmetrical activation functions, such as ReLU, that have a $R_{\geq 0}$ codomain.

This rises the need for a mapping from the original coordinate system to a local coordinate system. We empirically model a local coordinate system by providing images to the DL network and analyzing their extracted feature vectors in the representation space. We use two possible mappings:**Centering**: $x=L\left(e\right)=e-\bar{e}$,**PCA**: $x=L\left(e\right)={\bar{\Sigma }}^{-1/2}\left(e-\bar{e}\right)$, where $\bar{e}$ is the empirical mean and $\bar{\Sigma }$ is the empirical covariance matrix. The first option preserves the dimensionality n=K, while the second may operate a reduction by keeping the n<K biggest eigenvalues of $\bar{\Sigma }$. In the end, Tr=L∘E.

### 4.2. False Alarm Probability

The false alarm probability is usually defined as $P_{\mathrm{fa}}:=P\left(X\in D\right)$ where the random vector X models the feature vector of an original image. For the hypercone detector, under the weak assumption that X has an isotropic distribution (e.g., a Gaussian white distribution), this probability has a close form expression:(21)$$P_{\mathrm{fa}}=I_{\cos^{2}\left(\theta \right)}\left(\left(n-1\right)/2,1/2\right),$$
where I is the regularized Beta incomplete function.

This assumption does not hold for feature vectors extracted by the network. Even the empirical centering and whitening doesn’t guarantee a distribution *exactly* isotropic.

Instead, we prefer modifying the definition of the probability of false alarm. For a given image under scrutiny, the extracted features are seen as a fixed vector. It is the acceptance region which is random: the axis direction of the hypercone is a random vector A uniformly distributed on the unit hypersphere. It happens that this probability of false alarm has the exact same expression as (Equation 21). For a required false positive level, (Equation 21) is inverted so as to find the corresponding half angle value θ.

Another use of this equation is to compute the *p*-value: for given x and a, the *p*-value is the probability (Equation 21) computed for a cosine equalling $\cos\left(\theta \right)=|x^{⊤}a|/∥x∥$. It means that if we were drawing O(1/p) random secret keys A, on expectation, only one of them would give a normalized correlation bigger or equal than that cos(θ). The smaller the *p*-value is, the more convinced we are that a was the secret key used at the embedding.

### 4.3. The Objective Function and Imposed Constraints

A bad idea is to stick too closely to the watermarking described in Section 2.2. It amounts to first find $x_{m}$ (Equation 4) and then to use the back propagation to craft an image whose feature vector is as close as possible to $x_{m}$.

Watermarking in the feature space and crafting the watermarked image are better done jointly by defining:(22)$$\begin{array}{ccc} I_{w} & = & \arg\min_{\mathrm{Dist}(I,I_{o})\leq C}\mathrm{Loss}\left(I\right) \end{array}$$(23)$$\begin{array}{ccc} \mathrm{Loss}(I) & = & -R(\mathrm{Tr}(I)), \end{array}$$
and letting an adversarial sample mechanism solving this optimization problem. The distortion function Dist is for instance the Mean Square Error (expressed in dB as a PSNR). Note that with these notations, a watermark is detected in image I if Loss(I)<0.

While minimizing the loss and thus iteratively generating the watermarked image, additional constraints can be applied. For instance, the watermark can be attenuated selectively in the spatial domain by a perceptual mask to make it less perceivable. We use the Structural Similarity SSIM heatmap for that purpose. The watermarking is then performed by a simple gradient descent (see Algorithm 1).

### 4.4. Improving Robustness by Retraining

The introduction explains our motivation of benefiting of the inherent robustness of known classification DL networks in the very same way as it happened in image search. Retraining aims at increasing this robustness against typical attacks in watermarking applications. These are *image to image* transformations like translation, rotation, cropping or JPEG compression. Therefore building an augmented dataset is made easy by applying these transformations on the original training set. Note that watermarking is kept out of the retraining process. Our goal is to strengthen the invariance of the original classification network without any bias to the feature extraction method for watermarking. Later on, watermarking will naturally inherit this invariance.

The network is trained from scratch on a series of attacks of increasing difficulty. This is driven by the classification loss on the original network architecture (without the watermark feature extraction based on aggregation and whitening). For training, the ImageNet Large Scale Visual Recognition Challenge 2012 (ILSVRC2012) dataset [27] was used. It consists of 1.2 million images of 1000 classes. Attacks to the images were applied dynamically through a data augmentation pipeline making attacks of increasing difficulty as an input to the network (elaborated in Section 5.5).
**Algorithm 1:** Proposed watermarking algorithm.
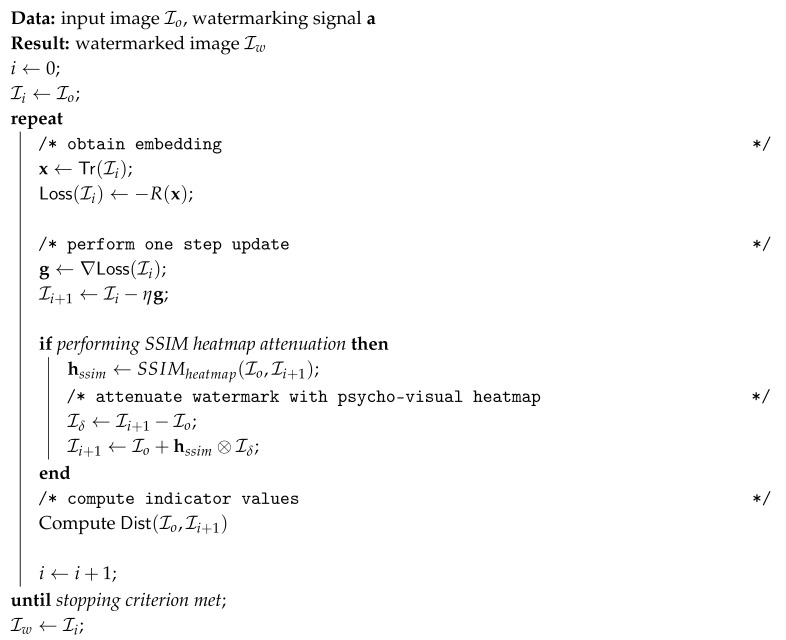


## 5. Experiments

The watermarking system is both architecture and layer agnostic, meaning that any layer can a priori be used for watermarking. This paper considers one specific architecture at different layers, exploring their goodness for watermarking by evaluating robustness and perceptual quality.

### 5.1. Experimental Protocol

The experimental part evaluates our proposed system based on the off-the-shelf *VGG19* [28] convolutional neural network pretrained on *ImageNet* (available at https://keras.io/applications/##vgg19). We perform all the experiments on photos from the test set P “professional” provided as part of the CLIC challenge [29]. This dataset consists of 117 professionally taken photos. The typical size is 2048×1350 pixels.

### 5.2. Image Quality

When the stopping criterion is expressed in terms of PSNR, the quality of the watermarked image is roughly acceptable above 40 dB. Figure 2 illustrates this with the reference Lenna image on its top row. The watermark signal in the image space looks like uniform noise when watermarking lower layers and gets more structured when processing deeper layers, as shown in Figure 3.

A psycho-visual attenuation as specified in Algorithm 1 improves the quality of the image. Our experiments used the structural similarity (SSIM) heatmap, but any other psycho-visual model (e.g., Butteraugli [30]) can be integrated transparently. Figure 2 shows that the watermark is less perceivable on the right column thanks to attenuation in flat regions (skin, hat, wall) for a given PSNR target. This almost does not hurt robustness $R(\mathrm{Tr}\left(I_{w}\right))$.

The downside is the increased number of steps (and thus time) necessary to obtain a watermark of the same robustness. Figure 4 shows a 3D representation of the watermarking process. The first two axis corresponds to the projection of $\mathrm{Tr}(I_{i})$ onto the basis (u,v) of Equation (Equation 6), while the third component accounts for the energy remaining in the complimentary space. The stopping criterion was met in 10 or 13 iterations, respectively without or with perceptual attenuation.

### 5.3. From One Layer to Another

Table 1 gives the $\log_{10}p$-value for the watermarked image and after a rotation of 5°. For the moment, no aggregation scheme is used. The first part of the table illustrates the behaviour when centering is used as a locality transform, thus retaining the original dimensionality. The second part of the table illustrates the behaviour with PCA with a reduction to 256 dimensions.

In general, watermarking at deeper layers offers more robustness. Each convolutional–pooling block adding some invariance to rotation and scaling, the best layer is indeed the first fully-connected layer.

### 5.4. Robustness without Aggregation

We evaluate the robustness of the first fully-connected *fc1* layer of VGG19 at the expense of a lower dimensionality (n=4096). From now on, the stopping criterion is defined as PSNR=42 dB and the probability of false alarm is fixed to $P_{\mathrm{fa}}=10^{-3}$. To define a baseline, we watermark the same images with the linear wavelet transform as described in Section 2.2 with the same distortion constraint.

Figure 5 illustrates the robustness against rotation. Wavelets drop to a detection rate of around 25% after a rotation of only 0.5° and it ceases to detect watermarks after a rotation of 1°. On contrary, the robustness of our technique smoothly degrades with the strength of the attack.

Figure 6 illustrates the robustness against cropping. The performances again smoothly degrade with the strength of the attack: 50% of watermark detection when cropped to 90% of their original content and continues to detect some watermarks down to a crop of 50%. The system outperforms the classic approach. This is not surprising as DWT was not designed for geometric invariance. Yet, it shows that accurate synchronization is of utmost importance with linear transforms (e.g., DCT, DWT, random projection), whereas it is far less stringent in the new approach.

On top of this, we observe that the system is robust to horizontal flips! We were not expecting this feature. After investigations, we provide the following explanation. Since horizontal flips are naturally and artificially occurring in the training dataset of ImageNet, the neural network seems to have learned a transform invariant to this.

Figure 7 illustrates the robustness against JPEG compression. DWT performs better and we argue that this is caused by the lower dimensionality (n=4k
*vs.*1M). The rule of thumb in watermarking is to spread the watermark to gain robustness against noise addition. This conflicts with the design of a domain invariant to geometric transformation.

### 5.5. Training Against Attacks

A new classification network is now trained from scratch. Its architecture is the same as the pretrained one used in the first part of our experiments, that is, VGG19. The ILSVRC2012 dataset that consists of 1.2 million images is augmented with attacked versions by the training pipeline. This pipeline introduces attacks of increasing strength. The network is trained for 1000 epochs which takes 3 weeks on an Nvidia GeForce GTX 1060 GPU with an Adam optimizer, a learning rate of 0.001, and a learning rate decay on plateau. The progression of the attacks is scheduled as follows:Initially, no attacks. Images come from the ILSVRC2012 dataset.After 5 epochs, horizontal flips and scaling within 90% and 110% are introduced.After 20 epochs, vertical flips and rotation of up to $3^{\circ }$ (both clockwise and anticlockwise) are introduced. Scaling is extended to the range from 70% to 130%.After 35 epochs, rotation is extended up to $\pm 5^{\circ }$.After 50 epochs, horizontal and vertical stretching up to 10% are introduced. Rotation is extended to $\pm 10^{\circ }$, scaling to the ranges from 50% to 150%.From epoch 150 on, attacks are performed with the following parameters: rotation up to $\pm 45^{\circ }$, scaling from 10% to 190%, horizontal and vertical stretching up to 50% and with horizontal and vertical flipping.

Training was performed in a standard classification setup with non-watermarked ImageNet images and is not related to the watermarking evaluation that follows.

Figure 8 illustrates the robustness to rotation of all the convolutional and pooling layers of the two VGG19 networks—the pretrained one (in black) and the one trained from scratch with progressive attacks (in red). Although there is a bit of overlapping for few layers, a clear separation is globally visible, indicating the better performances of the network trained with attacks. The best robustness is given by the pool4 layer of the network trained with attacks.

### 5.6. Robustness with Aggregation

Aggregation methods improve the performance of convolutional neural networks in classification and content based image retrieval tasks. Is this still the case for watermarking?

When evaluating against rotation, WELDON and RMAC aggregations improve the robustness compared to the feature extraction without aggregation. Figure 9 moreover shows that RMAC clearly outperforms WELDON for large angles. Its robustness is more stable, smoothly degrading as the rotation angle increases.

As for cropping, RMAC consistently outperforms WELDON and no aggregation at all as shows in Figure 10. For small cropping factors, the robustness of WELDON collapses more quickly than using no aggregation at all.

However, RMAC is not always the best aggregation. It is less robust than WELDON against JPEG compression (by a small amount—see Figure 11) and against a gamma correction (see Figure 12). Indeed, no aggregation at all seems to perform best against a gamma correction.

Note that Figure 9, Figure 10, Figure 11 and Figure 12 are not using the same scale on the *y*-axis. Overall, RMAC is the obvious choice especially against geometric attacks. Its relative weakness against valuemetric attacks is certainly due to the different $\ell_{2}$ normalizations. Appendix A shows some watermarked images with this option.

## 6. Conclusions

This work shows that watermarking features extracted from a classification network is feasible and relevant. Feasible because the framework of adversarial sample provides a way to create watermarked image of good quality. Relevant because the DL network provides a transformation which strikes a good trade-off between invariance to geometric attacks and robustness to valuemetric attacks.

This robustness is further improved thanks to mechanisms borrowed from other Computer Vision task like RMAC in image search. The training dataset plays also a big role. Augmentation with specific image transformations increases the invariance of the network although the training is driven by the classification performance and oblivious to watermarking.

From the application point of view, this study opens the door to a single image descriptor good for both image search and watermarking. In copy detection and copyright infringement applications, image search alone yields many false positives. A watermark detection would drastically decrease the number of false recognition cases.

The proposed approach has some drawbacks. First, embedding is roughly 20 times slower than detection—it is iterative and each iteration computes a gradient by back-propagation, whose complexity is the double of one inference. Reference [14] does not have this shortcoming because a generative network is in charge of the embedding. On the other hand, this generative network is not secret-keyed. Second, watermarking is not purely invariant to geometric transformation. This is fine in some applications where watermarked images face mild geometric attacks. For other applications, it will need a registration mechanism, yet that mechanism does not need to be perfectly accurate.

## Figures and Tables

**Figure 1 entropy-22-00198-f001:**
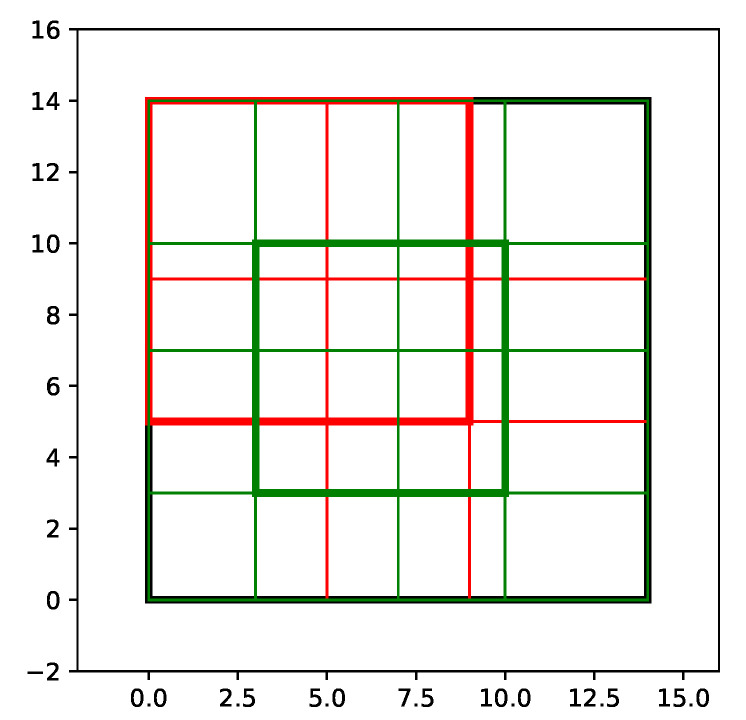
Three Regional Maximum Activation of Convolution (RMAC) levels and their corresponding pooling regions for a feature map of size 14×14×n. The first level, denoted in black and bolded, consist of only one region of size 14×14. The second level is denoted in red and consists of 4 regions (one of which is bolded for emphasis) of size 9×9. The third level, denoted in green, consists of 9 regions (one of which is again bolded for emphasis) of size 7×7 that are offset so that they have at least 40% overlap with the regions of the previous layer.

**Figure 2 entropy-22-00198-f002:**
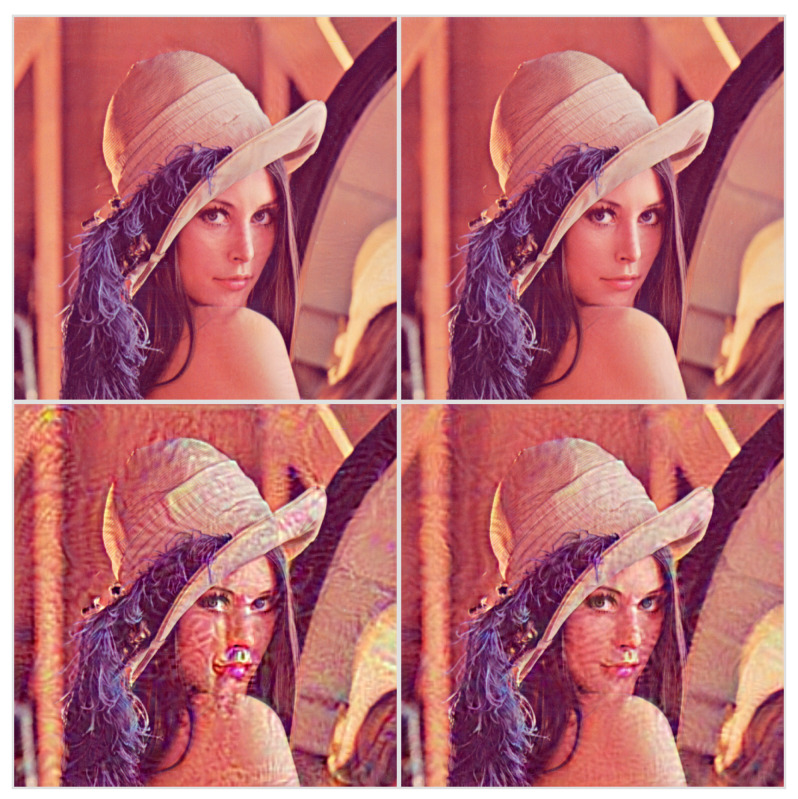
Top: PSNR=42 dB and $P_{fa}=10^{-8}$ without (left) and with (right) structural similarity attenuation. Bottom: ‘exaggerated’ watermark at PSNR=25 dB without (left) and with (right) structural similarity attenuation.

**Figure 3 entropy-22-00198-f003:**
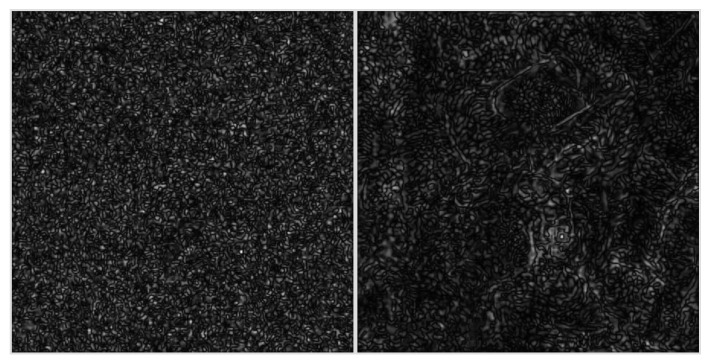
The watermark signal in the image space. Left - *block1_pool*: last pooling layer of the first convolutional block. Right - *fc1*: last fully-connected layer.

**Figure 4 entropy-22-00198-f004:**
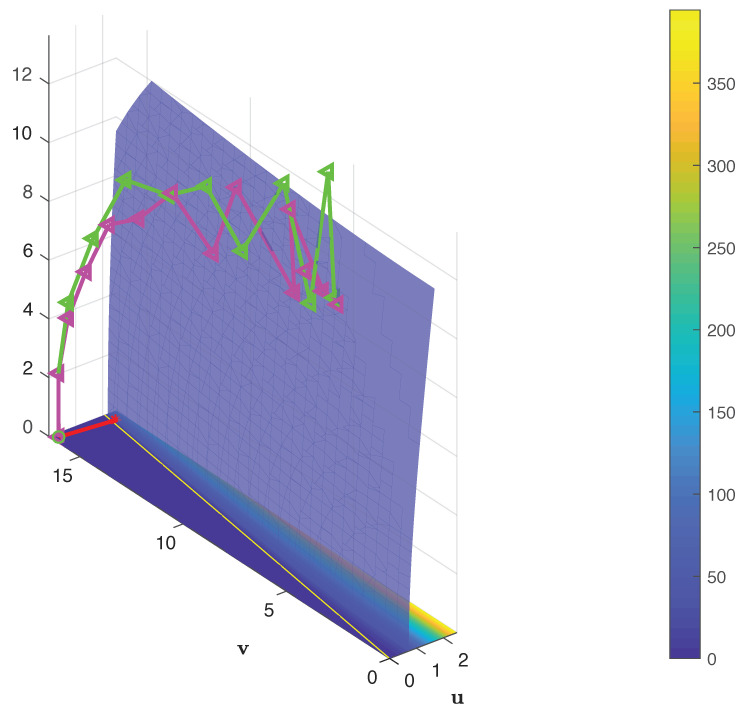
3D view of the iterations without (green) and with (magenta) the perceptual attenuation. The 3D shape shows the set {x:R(x)=c}. The red line shows the embedding of Section 2.2 providing the same robustness. The yellow line is the boundary of the hypercone. The heatmap of R(x) on plane (u,v).

**Figure 5 entropy-22-00198-f005:**
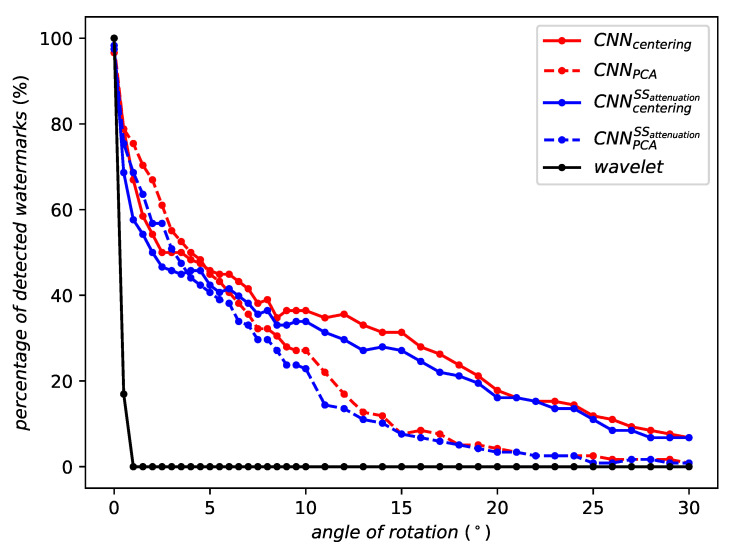
Robustness against rotation.

**Figure 6 entropy-22-00198-f006:**
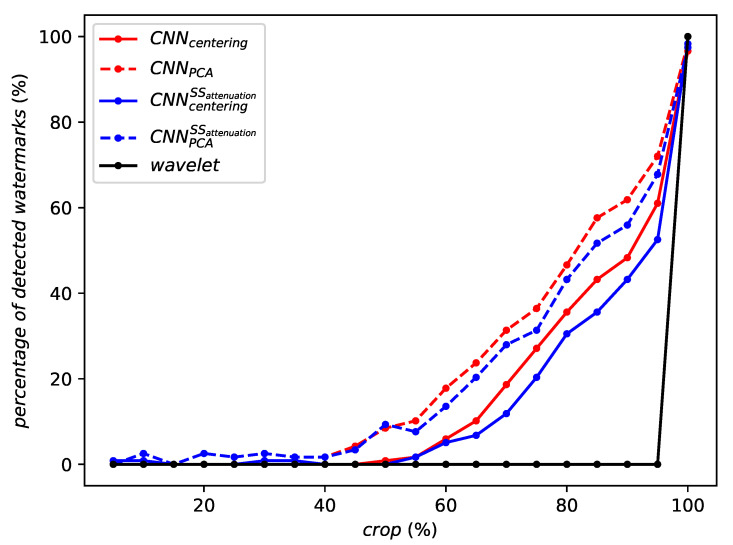
Robustness against cropping.

**Figure 7 entropy-22-00198-f007:**
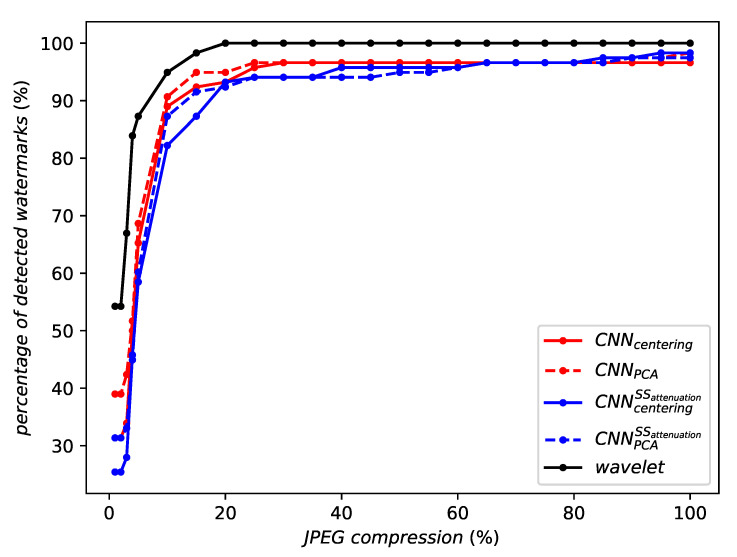
Robustness against JPEG compression.

**Figure 8 entropy-22-00198-f008:**
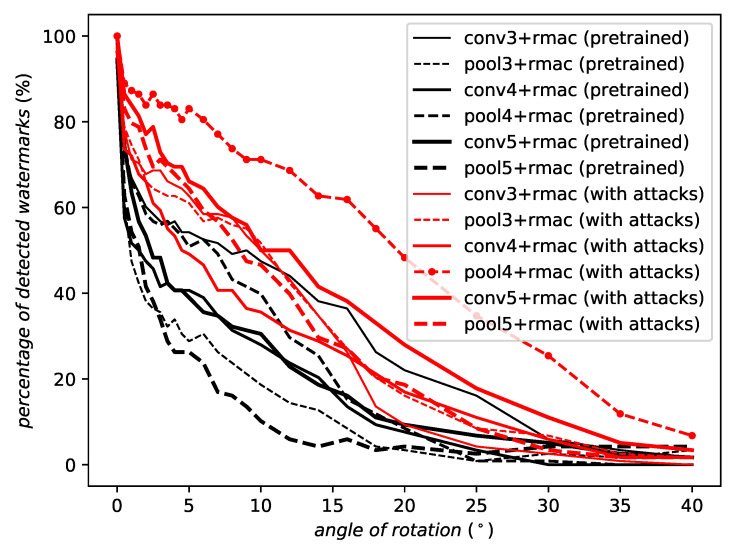
Robustness against rotation of different layers of the same architecture in a pretrained setup (black) and trained on a series of progressively harder attacks (red). Notation conv[i] indicates the output of the $i^{th}$ convolutional layer, pool[i] the output of the $i^{th}$ pooling layer of the VGG19 architecture, and +rmac indicates that the output is passed through an RMAC aggregation layer (Equation 10).

**Figure 9 entropy-22-00198-f009:**
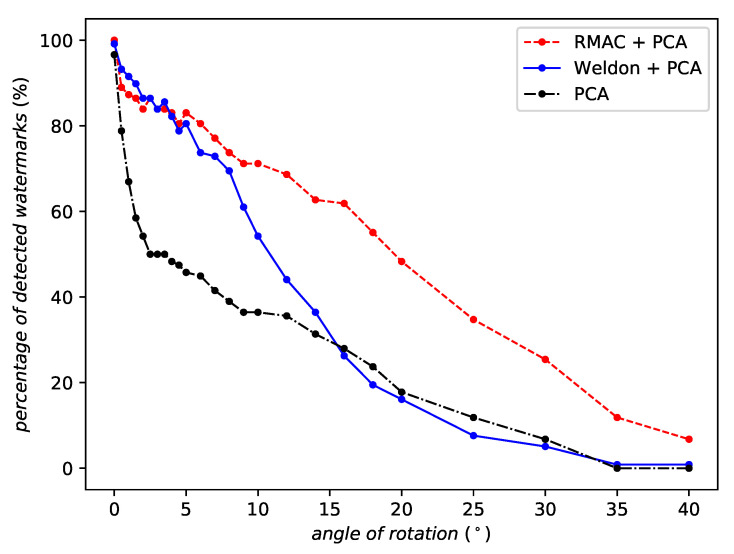
Robustness against rotation of the same network instance with different aggregation methods or no aggregation at all.

**Figure 10 entropy-22-00198-f010:**
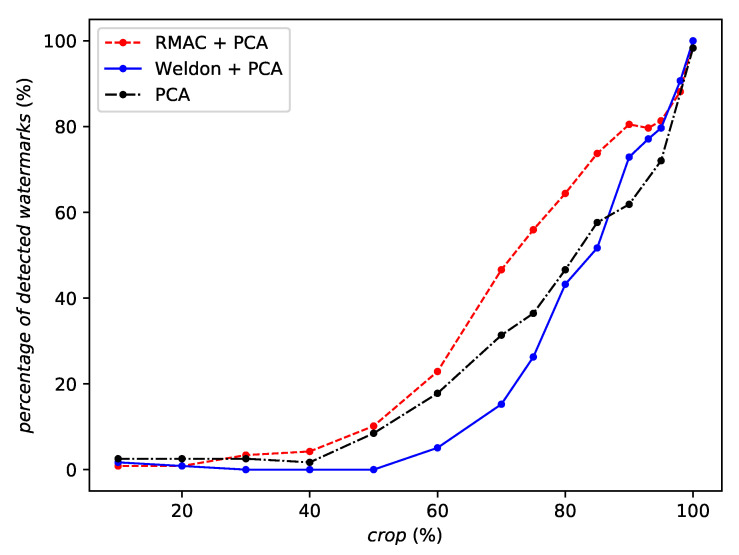
Robustness against cropping of the same network instance with different aggregation methods or no aggregation at all.

**Figure 11 entropy-22-00198-f011:**
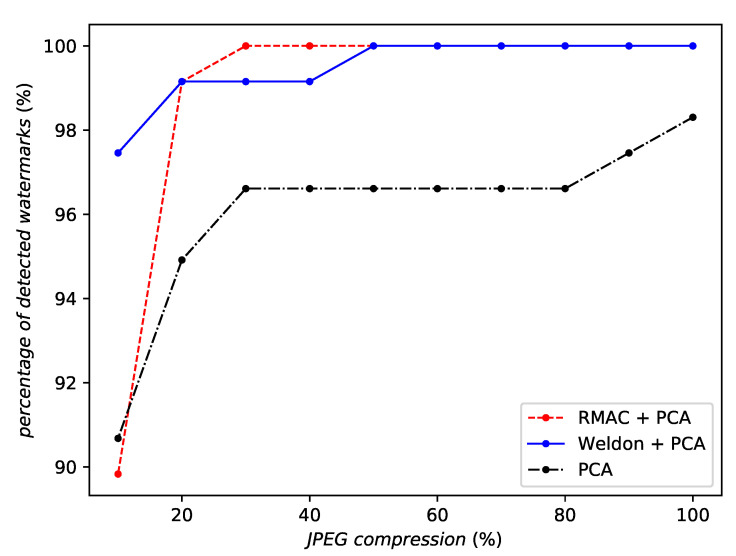
Robustness against JPEG compression of the same network instance with different aggregation methods or no aggregation at all.

**Figure 12 entropy-22-00198-f012:**
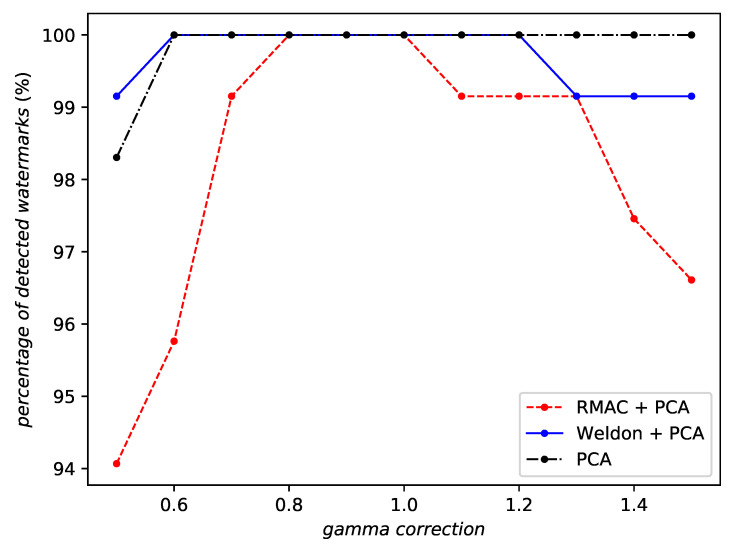
Robustness against gamma correction of the same network instance with different aggregation methods or no aggregation at all.

**Table 1 entropy-22-00198-t001:** Watermarking with different VGG19 layers with the same target $P_{fa}$ = 1× 10${}^{-6}$ and a stopping criterion of PSNR≤42 dB. (pool[i] being the output of the $i^{th}$ pooling layer and fc[i] being the output of the $i^{th}$ fully-connected layer of the VGG19 architecture).

Layer Name	Emb. Dim.	Average $\log_{10}p-\mathrm{Value}$ Marked Image	Average $\log_{10}p-\mathrm{Value}$ Rotation 5°
Original dimensionality, centering			
pool1	802 816	−64.49	−0.68
pool2	401 408	−58.32	−1.13
pool3	200 704	−52.25	−3.80
pool4	100 352	−21.31	−3.81
pool5	25 088	−4.95	−2.24
fc1	4 096	−5.27	−4.41
fc2	4 096	−4.11	−1.70
PCA to 256 dimensions			
pool1	802 816	−3.65	−0.93
pool2	401 408	−12.30	−1.36
pool3	200 704	−8.13	−3.20
pool4	100 352	−8.19	−2.42
pool5	25 088	−4.61	−2.41
fc1	4 096	−5.61	−3.20
fc2	4 096	−5.22	−1.69

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
