# Peer review of "Are Classification Deep Neural Networks Good for Blind Image Watermarking?"

_entropy, 2020, doi:10.3390/e22020198_

Round 1

Reviewer 1 Report

In this work the authors propose to use deep learning networks for zero-bit watermarking. The proposed scheme is well justified, and in general the method description is clear. My main concerns are:

- Probably it would be convenient to specify earlier that the proposed watermarking scheme is a zero-bit one.
- In Section 1.2 the authors mention [13] and [14], and they say that those methods "have a similar position than ours but they fail guaranteeing a probability of false alarm." Nevertheless, quantitative experimental results of those methods are not reported. In order to support their statement, it would be convenient to have a fair quantitative comparison.
- The novelty with respect to [16] should be highlighted. Similarly, the performance comparison with the authors' first approach should be paid more attention.
- First paragraph of Page 9, "Attacks to the images were applied dynamically:" maybe the authors can point the reader to Section 5.5 (Page 15), explaining that these dynamic characteristics are specified there.
- Surprisingly the authors only report the P_{fa} in Table 1. For Figs. 5 to 12 the authors only report the probability of detected watermarks (i.e., 1 - probability of missed detection;) please note that both false positives and false negatives should be considered.
- Concerning the "Training against attacks" described in Section 5.5, it is not clear if the attacks are only applied to watermarked contents, or also to non-watermarked ones. The scenarios resulting from both cases are pretty different, so it is relevant to make explicit that information.
- The notation blockn_pool, fcn, convn, pooln should be explained.

- Typos/writing:
* "stratgy" -> "strategy"
* "lower levels were less performing." Please check writing.
* "The third level, denoted in green, consist (sic) of 0 (??) regions"
* "continuous solution to obtained an image" -> "continuous solution to obtain an image."
* "a transformation which strike" -> "a transformation which strikes."

Author Response

Thank you for the comments. We have implemented most of your advices.

- "Probably it would be convenient to specify earlier that the proposed watermarking scheme is a zero-bit one."

We agree. This is now stated in the abstract.

- "In Section 1.2 the authors mention [13] and [14], and they say that those methods "have a similar position than ours but they fail guaranteeing a probability of false alarm." Nevertheless, quantitative experimental results of those methods are not reported. In order to support their statement, it would be convenient to have a fair quantitative comparison."

Above all, our paper is not about proposing a new watermarking technique. Our paper is a *study* aiming at answering the question of the title.
At first sight, [13,14] have already answered to this question. Yet, they have many pitfalls not acceptable for watermarking experts.

How can we have a fair comparison if these schemes do not fulfil the same requirements? They are not i) zero-bit, ii) not constrained on Pfa, iii) only dealing with small images,
iv) not containing any secret key, v) not controlling embedding distortion, vi) reporting indeed quite high distortion (35dB PSNR).
Note that in these papers, decoding on non-watermarked images (hypothesis H0 in our paper) is not even tested.
[14] has slightly investigated a zero-bit experiment where the detection rate is reported to be lower than 10% as soon as the quality factor of JPEG is lower than 90%.
Our Fig. 7 or Fig. 11 reports much better robustness, but again, it is not comparable (their images are smaller, their distortion bigger, etc).

We have detailed the differences in Sect. 1.2.

- "The novelty with respect to [16] should be highlighted."

Done in Sect. 1.2.

- "Similarly, the performance comparison with the authors' first approach should be paid more attention."

We have done this in Fig. 8, 9, 10, 11, and 12.

- "First paragraph of Page 9, "Attacks to the images were applied dynamically:" maybe the authors can point the reader to Section 5.5 (Page 15), explaining that these dynamic characteristics are specified there."

We agree. Done.

- "Surprisingly the authors only report the P_{fa} in Table 1. For Figs. 5 to 12 the authors only report the probability of detected watermarks (i.e., 1 - probability of missed detection;) please note that both false positives and false negatives should be considered."

This is because the Pfa is fixed for all Figs and tables throughout the paper (except Table 1).

- "Concerning the "Training against attacks" described in Section 5.5, it is not clear if the attacks are only applied to watermarked contents, or also to non-watermarked ones. The scenarios resulting from both cases are pretty different, so it is relevant to make explicit that information."

We agree. It is important to say that this training is independent of the watermarking layer.
Done in sect. 4.4.

- "The notation blockn_pool, fcn, convn, pooln should be explained."

We agree. We explain these terms in the caption of Fig. 8

- Typos/writing:

Thank you for spotting these. We made the corrections.

** Major addition in this new version:
- Sect. 1.2 has been augmented.
- Conclusion has been complemented with a discussion about the complexity.
- We have added an appendix displaying watermarked images and a discussion.

Reviewer 2 Report

In this paper, the authors proposed a novel watermarking algorithm. Relying on the technique of deep learning, the feature vector is extracted, instead of hand-crafted extraction. Furthermore, the adversarial sample is adopted to improve the performance of the watermarking algorithm, referring to as the imperceptibility. The structure of the paper is relevant, and the overall expression is acceptable. However, the only one issue should be carefully addressed. This paper is very similar to the WIFS paper [16] while the authors do not mainly address the differences between papers. In my humble opinion, the main contributions of this paper should be presented at the beginning, and emphasize the extended content compared to [16]. Besides, Abstract, Introduction, and Prior works, those sections are also very close to the content of [16]. I suggest the authors have to re-organize the whole paper, not simply copy the published paper.

Author Response

"... However, the only one issue should be carefully addressed. This paper is very similar to the WIFS paper [16] while the authors do not mainly address the differences between papers.
In my humble opinion, the main contributions of this paper should be presented at the beginning, and emphasize the extended content compared to [16].
Besides, Abstract, Introduction, and Prior works, those sections are also very close to the content of [16]. I suggest the authors have to re-organize the whole paper, not simply copy the published paper."

We completely understand your point. Here is our humble anwser, which is highly debattable. We refer to the Associated Editor whether this practice is acceptable.

Our point of view:
1) The conference paper was accepted to IEEE WIFS. This conference belongs to Electric Engineering where conference papers are short communication, the main work being the journal version. We believe that if we were submitting to IEEE TIFS (the journal related to IEEE WIFS), there would be no problem: i) we clearly cite the conference paper in the intro, ii) say a few words about the differences (We improve on that in this new version, see below**), iii) the differences are substantial. In this perspective, there is no point in hiding that some parts of the submission are copy-pasted from the conference paper... unless you find them badly written.

2) We submit this journal version to MDPI Entropy Special Issue (and not to IEEE TIFS) because we were invited to do so.

** Major addition in this new version:
- Sect. 1.2 has been augmented.
- Conclusion has been complemented with a discussion about the complexity.
- We have added an appendix displaying watermarked images and a discussion.

Reviewer 3 Report

1. Abstract

 - artefacts -> artfacts

 - "It also tests ~ " must be rewritten

 - authors have to consider upper cases for "Deep Neural Networks" and "Computer Vision"

2. Introduction

 - authors have to avoid "..." in the statements

3. Application to zero-bit watermarking

 - authors have to explain embedding and extracting algorithms, but only the embedding algorithm in Algorithm 1

4. Conclusion

 - "Feasible ~" and "Relevant ~" must be rewritten

Author Response

1. Abstract
- "artefacts -> artfacts"

artefact in English, artifact in American, but no artfact according to Cambridge dictionary.

- "It also tests ~ " must be rewritten

We agree. Done.

- "authors have to consider upper cases for "Deep Neural Networks" and "Computer Vision""

We agree. Done.

2. Introduction
- "authors have to avoid "..." in the statements"

We agree. Done.

3. Application to zero-bit watermarking
- "authors have to explain embedding and extracting algorithms, but only the embedding algorithm in Algorithm 1"

We disagree. We will not write Algorithm 2 just with a single line: Loss(I)<0 -> watermark! But we clarify this in the text. See Sect. 4.3

4. Conclusion
- ""Feasible ~" and "Relevant ~" must be rewritten"

We disagree. It is just a matter of style.

** Major addition in this new version:
- Sect. 1.2 has been augmented.
- Conclusion has been complemented with a discussion about the complexity.
- We have added an appendix displaying watermarked images and a discussion.

Round 2

Reviewer 1 Report

The authors have properly dealt with most of my comments. Just some considerations about two of them:
- Concerning the discussion included in Section 1.2 about [13] and [14], probably the authors should start by saying that those are multi-bit watermarking schemes, and then be a bit more careful in their subsequent discussion (e.g., "they use a single network at the detection side" -> "decoding side.")
- "This is because the Pfa is fixed for all Figs and tables throughout the paper (except Table 1)": please note that this is not explained (and the chosen P_{fa} is not specified) in Sections 5.4 to 5.6 (or in any other section of the paper, I think,) so it is not clear for the reader the value of P_{fa} in Figures 5 to 12. Furthermore, it would be interesting to know why the authors considered the same P_{fa} throughout the paper, but no in Table 1.

- Typo: "i.e." -> "i.e.,"

Author Response

"about [13] and [14], probably the authors should start by saying that those are multi-bit watermarking schemes..."

We agree. We now start by stating this.

"please note that this is not explained (and the chosen P_{fa} is not specified)"

You are right! Sorry for this. We added:

From now on, the stopping criterion is defined as $PSNR = 42$dB and the probability of false alarm is fixed to $\Pfp = 10^{-3}$.

Reviewer 2 Report

I have no further comments. 

Author Response

Few typos spotted.